# A rapid qualitative methods assessment and reporting tool for epidemic response as the outcome of a rapid review and expert consultation

Dong Dong[1], Sharon Abramowitz[2], Gustavo Corrêa Matta[3], Arlinda B. Moreno[4], Elysée Nouvet[5], Jeni Stolow[6], Caitlin Pilbeam[7], Shelley Lees[8], EK Yeoh[1], Nina Gobat[9], Tamara Giles-Vernick[10]*

**1** School of Public Health and Primary Care, Faculty of Medicine, The Chinese University of Hong Kong, Hong Kong, China, **2** Center for Global Health Science and Security, Georgetown University, Washington, D. C., United States of America, **3** Department Interdisciplinary Centre for Public Health Emergencies NIESP/CEE, Center for Data Integration and Knowledge for Health, FIOCRUZ - Oswaldo Cruz Foundation, Salvador, Brazil, **4** Department of Epidemiology and Quantitative Methods in Health, National School of Public Health, Oswaldo Cruz Foundation, Rio de Janeiro, Brazil, **5** School of Health Studies, Western University, London, Ontario, Canada, **6** Department of Social, Behavioral, and Population Sciences, School of Public Health and Tropical Medicine, Tulane University, New Orleans, Louisiana, United States of America, **7** ANU Medical School, School of Sociology, the Australian National University, Canberra, Australia, **8** Department of Global Health and Development, London School of Hygiene and Tropical Medicine, London, United Kingdom, **9** Country Readiness Strengthening, World Health Emergencies Program, World Health Organization, Geneva, Switzerland, **10** Anthropology & Ecology of Disease Emergence Unit, Department of Global Health, Institut Pasteur/Université Paris Cité, Paris, France

☯ These authors contributed equally to this work.

* tamara.giles-vernick@pasteur.fr

**Data Availability Statement:** All data used for this manuscript are available in the Supplementary files.

## Abstract

During the first year of the COVID-19 pandemic, the Methods Sub-Group of the WHO COVID-19 Social Science Research Roadmap Working Group conducted a rapid evidence review of rapid qualitative methods (RQMs) used during epidemics. The rapid review objectives were to (1) synthesize the development, implementation, and uses of RQMs, including the data collection tools, research questions, research capacities, analytical approaches, and strategies used to speed up data collection and analysis in their specific epidemic and institutional contexts; and (2) propose a tool for assessing and reporting RQMs in epidemics emergencies. The rapid review covered published RQMs used in articles and unpublished reports produced between 2015 and 2021 in five languages (English, Mandarin, French, Portuguese, and Spanish). We searched multiple databases in these five languages between December 2020 and January 31, 2021. Sources employing "rapid" (under 6 months from conception to reporting of results) qualitative methods for research related to epidemic emergencies were included. We included 126 published and unpublished sources, which were reviewed, coded, and classified by the research team. Intercoder reliability was found to be acceptable (Krippendorff's α = 0.709). We employed thematic analysis to identify categories characterizing RQMs in epidemic emergencies. The review protocol was registered at PROSPERO (no. CRD42020223283) and Research Registry (no. reviewregistry1044). We developed an assessment and reporting tool of 13 criteria in three

**Funding:** This work was funded by the World Health Organization (WHO APW-Emergency PO# 202651237). GM and ABM received funds in the form of a subcontract via Fiotec to create and pay language teams, develop the method employed for the rapid review, and conduct all data extraction.

**Competing interests:** Nina Gobat is an employee of the World Health Organization. The WHO funded this study.

domains, to document RQMs used in response to epidemic emergencies. These include **I. Design and Development** (i. time frame, ii. Training, iii. Applicability to other populations, iv. Applicability to low resource settings, v. community engagement, vi. Available resources, vii. Ethical approvals, viii. Vulnerability, ix. Tool selection); **II. Data Collection and Analysis** (x. concurrent data collection and analysis, xi. Targeted populations and recruitment procedures); **III. Restitution and Dissemination (**xii. Restitution and dissemination of findings, xiii. Impact). Our rapid review and evaluation found a wide range of feasible and highly effective tools, analytical approaches and timely operational insights and recommendations during epidemic emergencies.

## I. Introduction

Social science research sheds light on the social, cultural, political and economic dimensions of infectious disease outbreaks, their impacts and the impacts of public health and clinical responses; it has increasingly been used in global and local health emergencies to improve public health outcomes [1]. Governments, non-governmental organizations, research and global health funders, and scientific and medical networks recognize the need for social, cultural, political, and economic contextualization of response interventions and community engagement across broad publics and stakeholder groups [2]. In emergency contexts, rapid qualitative methods (RQMs) [3, 4]) generate rapid, real-time analyses that show how individuals and communities understand and respond to epidemic disease, interventions, and interactions with healthcare systems [5–7].

RQMs were of substantial interest before 2020, exemplified by a 2017 systematic review of English language sources identifying RQM purposes, strengths and limitations in complex health emergencies [3]. Nevertheless, the COVID-19 pandemic catalysed substantial expansion in RQM use, as well as innovative qualitative data collection methods via email, journaling, online platforms, and social media [4, 8–10]. Improvements in qualitative research capacity, partnerships, and use have also been reported during the pandemic, even as social distancing limited researcher access to community and healthcare facility research sites. Yet this increased RQM use has raised new questions. There is little understood about RQM deployment, impact, and reporting for non-English language sources coverage global regions where English publication is not a given. More broadly, with more widespread RQM use, we need tools to evaluate RQM quality in epidemic emergences and to strengthen study reporting.

This article, then, expands this prior research to review RQMs in English, French, Spanish, Portuguese, and Mandarin Chinese languages and to include new geographical regions (Asia, Latin America) and new or under-recognized RQMs. Based on this rapid review and analysis, we built an assessment and reporting tool instrument to guide the development and implementation and to evaluate the quality of RQM studies conducted in epidemic emergencies. It supports greater transparency in developing, implementing and reporting RQM study results. This RQM assessment and reporting tool can thus assist a wide range of users, including program designers, funders, researchers, evaluators, and peer reviewers.

## II. Methods

The Methods Subgroup of the WHO COVID-19 Social Science Research Roadmap Working Group conducted this rapid review from October 2020 to February 2021 in a process that

included serial parallel review processes and 15 virtual research and coordination meetings. We investigated the framing and application of rapid qualitative methods during recent epidemics, with an emphasis on the COVID-19 pandemic. We identified data collection tools, research questions, research capacities, analytical approaches, and strategies that were employed to speed up data collection and analysis in their epidemic and institutional contexts. The review objectives were as follows:

1. To synthesize the implementation and uses of RQMs in epidemic emergencies, their content, contexts in which they are used, research questions to which they are applied, research capacities required, and how they speed up data collection and analysis

2. To propose an assessment tool to set standards for RQM implementation and reporting in epidemic emergencies.

## A. Approach

In October 2020, a six-person Core Group with proficiency in the five languages developed the rapid review protocol (PROSPERO no. CRD42020223283; Research Registry no. reviewregistry1044). We prioritized inclusion of global languages, research traditions, and geographic locations in our methodology. The full Methods Subgroup was comprised of 11 qualitative researchers with rapid qualitative research expertise in infectious disease outbreak settings.

Phase 1 was conducted by three language teams, consisting of 2–5 members. There were some minor variations in search strategy. Team 1 focused on English and Mandarin language sources, with English-language researchers beginning their search with a 2017 start date so as not to replicate a previous systematic review [3]. Team 2 focused on French language sources. Team 1 and Team 2 set no limits on geographic boundaries. Team 3 researched Spanish and Portuguese language publications in the South/Central American region, noting that there had been no prior reviews of RQMs in epidemics conducted in South/Central America and that a full search for the entire world would produce more sources than could be evaluated during a rapid review. All other searches were set to cover 2015 to the present to include major recent epidemic emergencies, namely Ebola and Zika) (See S1 File for search strings).

The teams worked independently on translation of search terms, database selection, and manual inclusions, and met weekly to ensure coherence and consistency.

## B. Search databases and strategy

**Search strategy.** Initially, six-person Core Group tried to use English search terms from a previously published study [3] but could not successfully replicate the resource extraction because we did not use all the same databases, notably for our non-English language searches, and because the prior study offered a "sample search strategy" that was not replicable across all databases due to their varied constructions. We then sought input from expert peers in the larger Methods Subgroup to develop new common search terms, translated into five languages. After initial testing of the search terms in English and translation and testing to other languages, all research teams implemented the search using a shared translated search thread in 3–5 database searches. Table 1 presents the search languages, databases, date ranges, and regions covered by the rapid review. To ensure inclusion of recent and unpublished methodologies, we used the same search terms and inclusion/exclusion criteria to review and manually add contributions from the WHO Social Science Community of Practice, Global Health Cluster, various social sciences networks (Medical Anthropology Switzerland, Medical Anthropology at Home, and LinkedIn). We also searched professional and technical websites several site (ex. Relief Web, as well as UN, government publications, and NGO websites).

**Table 1. Languages, search databases, websites, date ranges, and regions covered.**

| Language | Databases | Date Range of Search | Regional limitations |
|---|---|---|---|
| English | PubMed, Medline, EMBASE, Web of Science | 2017*-2021 | All regions |
| French | PubMed, HORIZON, SUDOC | 2015–2021 | All regions |
| Mandarin | SinoMed, CNKI, Airiti Library | 2015–2021 | All regions |
| Spanish & Portuguese (LAS) | LILACS, Web of Science, BVS—Health Library in Health (PAHO/Brazilian Ministry of Health), Scopus, Scielo, PubMed | 2015–2021 | Latin America ⱡ |
| | **Unpublished Reports from Community** | | |
| English & French | WHO Social Science Community of Practice, Global Health Cluster | 2021 | All regions |
| | **Social Networking sites** | | |
| English & French | Medical Anthropology Switzerland, Medical Anthropology at Home, LinkedIn | 2021 | All regions |
| | **Websites** | | |
| English & French | Relief Web and UN, government publication, and NGO websites | 2015–2021 | All regions |

*The English language search collected sources from 2017 to 2021 to avoid duplication of sources collected in a published systematic review.

ⱡ Only Latin America-specific publications were selected for this search because no prior rapid or systematic assessment had been conducted on RQM research emerging from LA experiences with epidemics, despite substantial research conducted there.

## C. Selection criteria (inclusion/exclusion) and process

To determine selection criteria, each language team (led by Core Group members) conducted a first-round review by title and type of article, a second-round review of abstracts from sources, and a third-round review of full-text documents. Teams were divided into double-blind peer-review pairs. Conflicts were resolved through team-based deliberation. For the inclusion criteria, included sources needed to match all criteria. Excluded sources matched one or more exclusion criteria. Table 2 lists inclusion and exclusion criteria. The date of last inclusion of sources was January 31, 2021.

Relevant start dates were determined in exchanges with our funding. English language source search began in 2017, following [3], whereas non-English language source search began in 2015.

Extracted data included full citation and detailed textual descriptions of the key methodological dimensions in the papers, including study objectives, tool(s) used, time frame of the study (initially designated six months or less from study conception to reporting, but subsequently, delineated as data collection completion because sources lacked clarity about study duration), training required to conduct the study, recruitment procedures, ease of data collection and analysis, community engagement and reporting of results back to communities and stakeholders, and impact of the study. To extract data from the included articles, teams used a pre-defined template developed by the six Core Group members (S2 File). The Core Group developed a 25-point template to extract data on the creation, contexts, implementation, and impact of RQMs in epidemics. Contextually, the numerous demands for RQMs in the COVID-19 pandemic also influenced Core Group attention to specific data extraction points, for instance, a tool's capacity to identify social vulnerabilities or to be repurposed in other epidemic settings. A single reviewer extracted the data; the language team leader read the source and reviewed the data for accuracy and completeness. Again, disagreements about coding were resolved by discussion within individual teams and within the core group to reach consensus.

In Phase 2, the 11-member Methods Subgroup analysed and reviewed code sheets, leading to iterative refinements in coding and classification (S3 File). A second round of coding was

**Table 2. Inclusion and exclusion criteria.**

| | |
|---|---|
| **Inclusion criteria** | • Described by authors of the potential source as "rapid" relative to the urgency of the health situation; OR full implementation (training, data collection, analysis, and results) produced within six months<br>• Focused on infectious diseases with epidemic potential (including during natural disasters)<br>• Used an ethnographic or qualitative research approach<br>• Investigation conducted in the context of an outbreak, epidemic, or pandemic<br>• Investigation conducted between 2017–2021 (English sources) or 2015–2021 (Mandarin, French, Portuguese, Spanish sources)*<br>• Sources that are standard peer-reviewed journal publications and research reports, unpublished field and methodological notes, or grey literature reports.<br>• Sources in English, French, or Mandarin focusing on rapid research conducted anywhere in world, and sources in Spanish and Portuguese languages addressing research conducted only in Latin American countries |
| **Exclusion criteria** | • Addressed non-human populations only<br>• Used quantitative methods only<br>• Did not address a context of high outbreak or epidemic disease transmission<br>• Conference presentation containing only title and abstract<br>• Source did not report and analyze primary research (commentary, letter, review)<br>• Addressed non-communicable diseases, including mental health disease, substance abuse, or addiction, in non-emergency setting<br>• Published prior to 2017 (English sources) or prior to 2015 (Mandarin, French, Spanish, Portuguese sources)<br>• For Spanish/Portuguese sources, addressed countries outside of Latin America |

conducted to clarify analysis regarding the following coding criteria: tool selection, training, rigor and replicability of research methods, analysis techniques, and study design. In the second round of coding, expert researchers reviewed abstracts and first coding for each source and read the entire publication or consulted the original language team for additional information.

In our final round of extraction, we entered all data into MS Excel to identify and organize key features of selected RQMs. This tabulation of features included the range of objectives, timing of studies, types of epidemics, peer-reviewed publication or grey literature, locations, tools and associated measures to enhance rapidity, analytical approaches, and impact of the study, among other features. These assessments provided the basis for synthesizing our global comparison of all included studies. To ensure that all Methods Subgroup members coded in a consistent and coherent fashion, we evaluated intercoder reliability (ICR) by using an online tool, ReCal OIR ("Reliability Calculator for Ordinal, Interval, and Ratio data") [11]. Divided into five teams, the 11 Subgroup members coded a sample of 11 English-language sources, which converted to 99 analysis units. Intercoder reliability (ICR) was calculated by Krippendorff's alpha. The result was $\alpha = 0.709$, which exceeds the lowest acceptable reliability ($\alpha = 0.667$) at the 0.005 level of statistical significance [12].

## D. Risk of bias and quality assessment of included studies

We used several methodologies to reduce bias, but a key limitation was the lack of sufficient information in selected publications to evaluate quality and risk of bias. This lack of sufficient information supports the need to develop a framework for evaluating and reporting of rapid qualitative investigations.

## E. Data analysis and synthesis

We employed thematic analysis to identify the categories characterizing RQMs in epidemic emergencies, to assess RQM adaptability to specific contexts and yield robust qualitative evidence, and to evaluate impact. Notably, many identified RQMs lacked sufficient descriptive detail regarding methods (see Results). These thematic categories thus constituted key indicators of the quality of RQM studies in epidemic emergencies. In the COVID-19 pandemic context, the Methods Subgroup sought to extend our analysis and apply our expertise so that RQM users could draw from our rapid review and to design and implement RQMs in epidemic emergencies, to evaluate their quality, and to engage in transparent reporting. To that end, the Methods Subgroup grouped and reorganized identified themes into chronological order and subsequently, through extended collective reflection and discussion, elaborated the 13 key criteria and three domains of our RQM assessment and reporting tool.

A summary of the 126 included sources of published and gray literature appears in S5 File.

## III. Results

The searches led to 8405 documents included in the first review round, and 126 sources in the final sample. Fig 1 presents details of the selection of included sources. A breakdown of PRISMA flow diagrams by language group is available in S4 File.

### A. Overview of sources

The multi-lingual search strategy yielded 126 sources that covered a broad geographic, language, and health emergency diversity. All sources were based on primary research, of which 92% (114 of 126) appeared in peer-reviewed publications; the remaining 7% were grey literature. Of the included sources, 45% were in English language, 38% were in Mandarin Chinese, 10% were in Spanish and Portuguese from South/Central American (SA) countries, and 6% were in French. As Table 3 demonstrates, nearly half of all articles reviewed conducted RQMs in Asia (49%). Of the remaining articles, the distribution was as follows: South/Central America (18%), Africa (13%), North America (8%), Europe (8%), Middle East (7%), and Australia (2%). Rapid qualitative research was most often conducted in LMICs [13]. Of the 41 countries represented in the included studies, 73% (n = 30) of the countries represented were LMICs; 27% (n = 11) were HICs. These 41 countries were: Argentina, Australia, Bangladesh, Benin, Brazil, Burkina Faso, Cambodia, Cameroon, Canada, Chad, China (including Taiwan), Colombia, Cuba, France, Guinea, India, Indonesia, Iran, Iraq, Jordan, Lebanon, Liberia, Malawi, Nigeria, Oman, Pakistan, Paraguay, Peru, Saudi Arabia, Senegal, Sierra Leone, Singapore, South Korea, South Sudan, Sweden, Thailand, Turkey, Uganda, UK, USA, Yemen. Eight studies were coded as «Other» (multi-country).

Twelve percent (n = 15) of included sources characterized their research as "rapid". Included sources rarely used the terms "rapid qualitative methodology" and "rapid qualitative assessment,", or indicated that research activities were completed in less than six months. Eighty-eight percent of all studies were initiated and completed over a range of several days to four months (see Table 3).

Among the included sources, Mandarin Chinese and English-language sources used RQMs to conduct research in Asia (see Fig 2). French and English-language RQMs were used in Europe, Africa, and North America. French and English-language RQMs were used in Europe and Africa. English-language only RQMs were used in Australia and North America. English, French, Spanish, and Portuguese were used in multi-country studies. Spanish, Portuguese and English-language were used in South/Central America.

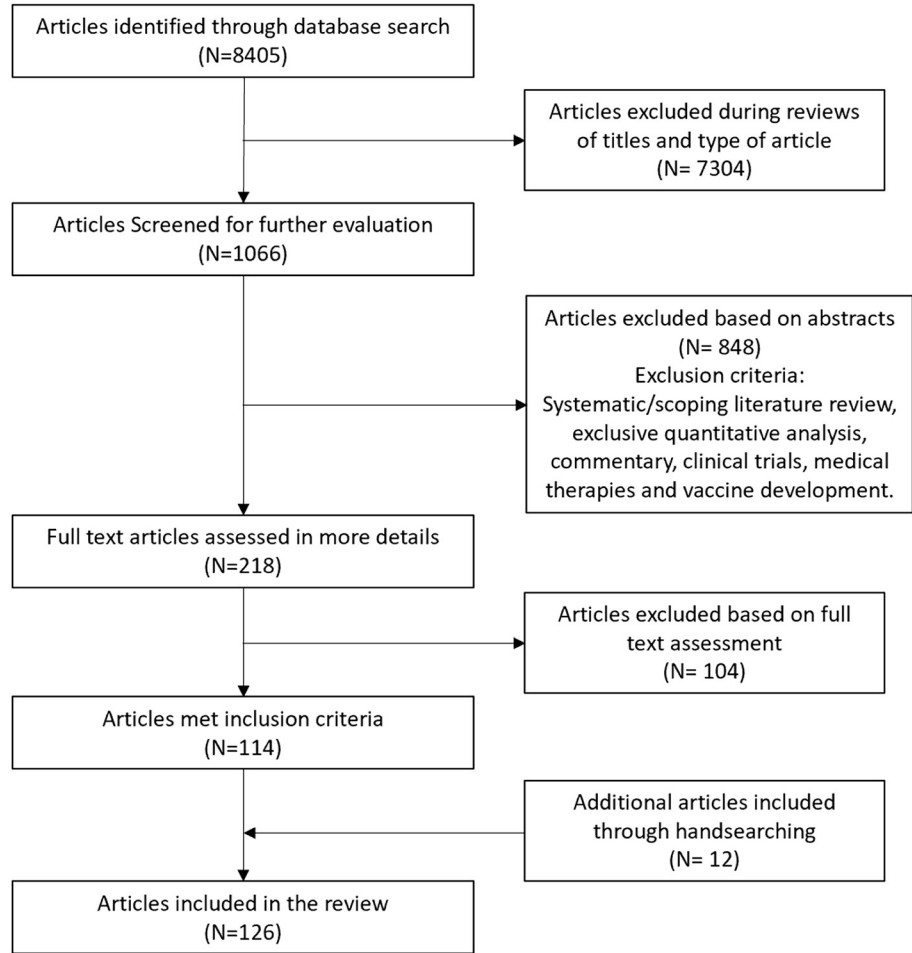

**Fig 1. PRISMA flow diagram showing steps for selection of included sources.**

Seventy-four percent of the included studies were conducted in 2020, which led to a large proportion of investigations addressing COVID-19. Sixty-seven percent of these studies specifically addressed the COVID-19 epidemic; 10% focused on Ebola, whereas 7% dealt with Zika, and 16% addressed other diseases Several articles addressed more than one public health emergency, and each public health emergency was counted as an individual case, resulting in a total greater than 100% (Table 3). Several sources could not be categorized geographically because they were global or multi-regional in scope. Although the Methods Subgroup attempted to provide categorization of sources by geographical scale (local, national, regional), our included sources did not fit neatly into this scheme. Indeed, several studies were multi-sited, multi-country and/or trans-continental [14–17].

Publications covered a wide range of outbreaks. In the English language sources, diseases included COVID-19 (>50% of articles), Ebola, Zika, Cholera, Middle East Respiratory Syndrome (MERS-CoV), natural disasters, zoonotic diseases, diphtheria, febrile illnesses, Hand, foot and mouth disease (in this case, considered an emergency), diarrheal disease, and combined diseases. In the South/Central American literature, approximately half of all sources addressed COVID-19, and the remainder addressed arboviruses (Yellow fever, Dengue and Zika), with Zika predominating. In the Mandarin literature, most sources reported on

**Table 3. Evidence review: Languages, regions, year conducted, duration, and health emergency.**

| Languages | N | % | Health Emergency | N | % |
|---|---|---|---|---|---|
| English | 57 | 45 | COVID-19 | 85 | 67 |
| French | 8 | 6 | | | |
| Mandarin | 48 | 38 | Ebola | 12 | 10 |
| Portuguese/Spanish (LAS) | 13 | 10 | Zika | 9 | 7 |
| **Regional Distribution*** | **N** | **%** | Natural Disasters | 3 | 2 |
| Asia | 62 | 49 | Cholera | 3 | 2 |
| South/Central America | 23 | 18 | MERS | 3 | 2 |
| Africa | 16 | 13 | Dengue | 2 | 2 |
| North America | 8 | 6 | Hand, Foot, and Mouth Disease | 1 | 1 |
| Europe | 8 | 6 | Diarrheal Disease | 1 | 1 |
| Middle East | 7 | 6 | H7N9 | 1 | 1 |
| Australia | 2 | 2 | Zoonotic Diseases | 1 | 1 |
| **Year** | **N** | **%** | Diphtheria | 1 | 1 |
| 2020 | 93 | 74 | Other (*multiple diseases, infectious diseases in general, public health emergencies in general*) | 4 | 3 |

| | | | | English | French | Mandarin | Spanish/Portuguese | N |
|---|---|---|---|---|---|---|---|---|
| 2019 | 9 | 7 | COVID-19 | 52.6% | 50.0% | 93.8% | 46.2% | 85 |
| 2018 | 8 | 6 | Ebola | 12.3 | 50.0 | | 7.7 | |
| 2017 | 13 | 10 | Zika | 8.8 | | | 30.8 | 12 |
| 2016 | 1 | 1 | Natural Disasters | 5.3 | | | | 9 |
| 2015 | 2 | 2 | Cholera | 5.3 | | | | 3 |
| **Timeframe of Research** | **N** | **%** | MERS | 5.3 | | | | 3 |
| < 1 week | 18 | 14 | Dengue | | | | 15.4 | 3 |
| 1–2 weeks | 15 | 12 | Hand, Foot, and Mouth Disease | 1.8 | | | | 2 |
| 2 weeks– 1 month | 19 | 15 | Diarrheal Disease | 1.8 | | | | 1 |
| 1–2 months | 23 | 18 | H7N9 | | | 2.1 | | 1 |
| 2–3 months | 21 | 17 | Zoonotic Diseases | 1.8 | | | | 1 |
| 3–4 months | 17 | 14 | Diphtheria | 1.8 | | | | 1 |
| 4–5 months | 3 | 2 | Other | 3.5 | | 4.2 | | 4 |
| 5–6 months | 8 | 6 | N | 57 | 8 | 48 | 13 | 126 |
| ≥ 6 months | 1 | 1 | | | | | | |

COVID-19 (45/48 articles, with 32% conducted in Wuhan and Hubei provinces, the initial COVID-19 epicentre); o*thers addressed H7N9 (1), infectious diseases in general (1) and public health emergencies in general (1)*. French sources addressed COVID-19 and successive Ebola epidemics in West and Central Africa.

## B. Methods used for RQMs

Our initial intent was to document the landscape of rapid qualitative methods used during public health emergencies. Among the 126 articles extracted, the most used RQMs were key informant interviews (KIIs) (75%), Focus Group Discussions (FGDs) (19%), Digital/Online research (14%), Ethnographic research (12%), and Surveys or questionnaires (8%), and Document analysis (7%). Nearly half of all sources used multiple methods to collect data. Fifty-seven percent (71) sources used 1 rapid qualitative method; 34% (42) used 2; 7% (9) used 3; and 2% (2) used 4 RQMs; three articles did not provide sufficient detail to code.

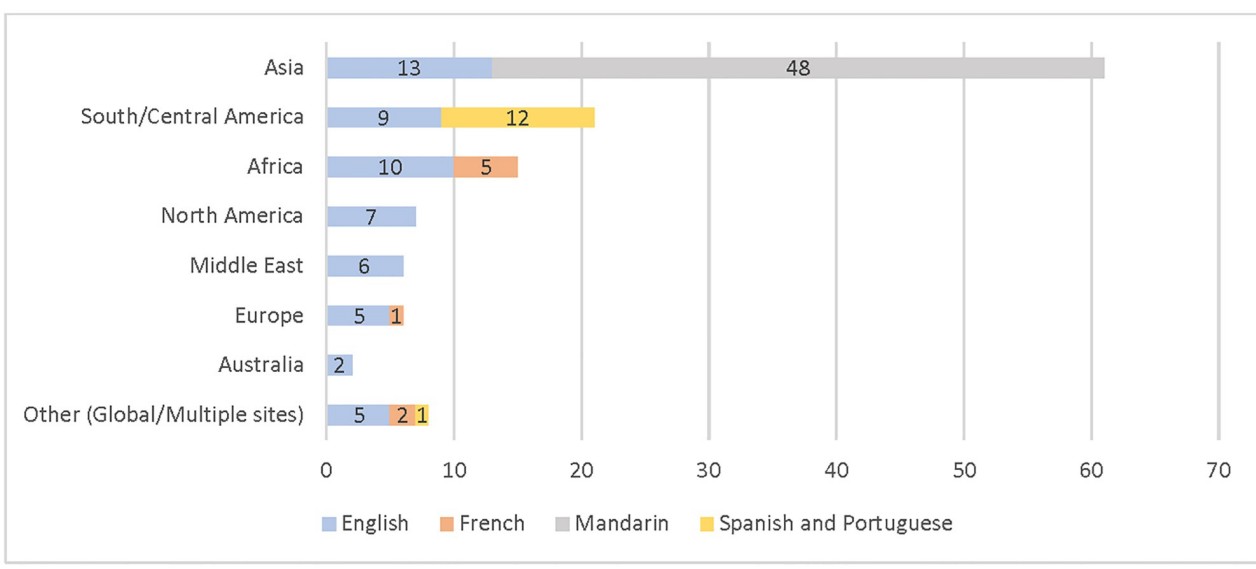

**Fig 2. Distribution of regions by RQM language.**

Table 4 shows the six most frequently used RQMs used by researchers. Descriptions of methods varied substantially and could include:

- specific *methods* employed (daily logs, surveys, focus group discussions) descriptions of the *methodologies* deployed (for instance, "ethnographic research", which uses specific tools of participant-observation, informal one-to-one discussions, and informal group discussions)

**Table 4. Most frequently used methods in RQMs.**

| Methods | RQM | N | % |
|---|---|---|---|
| • Key informant interviews | Key informant interviews | 95 | 75 |
| • Focus group discussions | Focus group discussions | 24 | 19 |
| • Ethnographic research | Digital/online | 18 | 14 |
| • Participant-observation | Ethnographic research | 15 | 12 |
| • Field observation | Survey/questionnaire | 10 | 8 |
| • Informal discussions | Document analysis | 9 | 7 |
| • Community meetings | | | |
| • Community mapping | | | |
| • Rapid clinical ethnography (RAPICE) | | | |
| • Surveys | | | |
| • Questionnaires | | | |
| • Document analysis | | | |
| • Case study | | | |
| • Rapid implementation action research | | | |
| • Oral history | | | |
| • Official data review | | | |
| • Daily log | | | |
| • Digital/online | | | |
| • Participatory action research | | | |
| • Delphi technique | | | |
| • Collective subject discourse | | | |

- depictions of the *overarching theoretical approach* (for instance, Participatory Action Research, Collective Subject Discourse Theory) informing the methods used, rather than specific methods.

Some RQMs were broad and diverse. Digital/Online methods, for instance, included content analysis of social media, real-time observation, participation, and analysis of text-based platforms, or web-conference-based interviews or focus groups. Document analysis referred to reviews of government documents, archival materials, diaries or logs, or wide-ranging policy reviews.

Although most sources described their qualitative data collection methods, only 17% (n = 21) published specific details of the methods used [18, 19], such as interview or focus group guides. Publishing such details not only allows readers to gain insight into the types and framing of questions asked, but it can also save time by facilitating speedy tool selection and adoption and can help promote replicability of the study design [6, 20].

Some RQMs, particularly those conducted during the COVID-19 pandemic, relied on remote data collection tools (telephone, online videoconferencing platforms). In some parts of the world with good connectivity, remote data collection worked well [21], but elsewhere, remote data collection was hampered by unreliable online connections [19] notably in certain LMICs with poor internet service. Online data collection can introduce sampling bias, favoring those with online access [21–24].

## C. Analytical approaches

Just as the methods were heterogeneous, so too were data analysis approaches. The most frequently reported data analysis strategies included content analysis, narrative review, qualitative analysis, thematic analysis, lexicographic analysis, word cloud techniques, similitude analysis, and Colaizzi analysis.

We were unable to fully identify, classify, and rank according to preference different analytical approaches. Included sources, including peer-reviewed ones, described their analytical approaches in such a way that they either lacked clear definition of their data analysis approaches or deployed analytical approaches that overlapped substantially, making it nearly impossible to distinguish one method from another.

## D. Regional variation in research design

We found that different regions have different legacies and traditions for conducting social science research in epidemics. Regional differences were observed in the theoretical and methodological choices and presentations reported in publications, as well as in scale, staffing, approach, and preferred methodological tactics.

Below are some examples of regional variation in theoretical and methodological traditions, and most frequently used "go-to" methods.

- Research published in English tended to be most closely associated with achieving programmatic agendas. They employed a range of qualitative collection tools, included interviews, focus group discussions, and ethnographic research (participant observation). They also used specialized methodologies (Rapid Clinical Ethnography (RAPICE), Delphi consensus processes), and highly structured processes (structured and semi-structured interviews and checklists).

- South/Central American researchers utilized research methodologies grounded in South/Central American experience: like qualitative, participatory action research based on the

three phases of Paulo Freire's research itinerary: Thematic Investigation; Coding and Decoding; and Critical Unveiling [18, 25].

- Mandarin-language studies tended to involve small-sample, interview-focused research in health care settings (notably hospitals) and were conducted by health care workers. These studies were largely published in Mandarin-language medical or nursing journals.

- French-language studies tended to highlight ethnographic research and informal discussions and used semi-structured interviews and focus group discussions.

## E. Speeding up research

Per our criteria, "rapid" qualitative research had a duration of less than six months from inception to completion. Extracted sources documented strategies to accelerate research, including adaptations to data collection, analysis, and reporting processes. These modifications were made either explicitly or implicitly. We were not able to assess the impact of accelerated strategies on study research rigor. Table 5 shows 10 strategies used to accelerate research in extracted studies.

## F. Ethics

Approximately 1/3 of all extracted research studies (n = 45, 35.7%) reported undergoing human subjects research ethics review processes, received approval from institutions, and obtained informed consent from participants. Forty-five sources reported that they received

**Table 5. Ten observed strategies for making RQMs "Rapid".**

|   | Strategy | Description |
|---|---|---|
| 1 | *Large research teams* | Larger research teams with multiple experienced senior researchers and research assistants could collect and analyze more data, and faster [6, 26, 27] |
| 2 | *Professionals and Key Stakeholders as Respondents* | Research was conducted with more easily identifiable and easy-to-reach populations, such as staff, key stakeholders, and other personnel from state agencies, NGOs, hospitals, or humanitarian organizations, rather than with at-risk populations or recipients of assistance or care [28–30] |
| 3 | *Standardized data collection methods and protocols* | Standardized rapid qualitative assessment guidance tools, such as RAPICE, accelerated data collection and analysis [7, 31] |
| 4 | *Precision data collection* | Standard qualitative data collection techniques were adapted to focus with precision on participant priorities, concerns or practices ("mini-focus groups", extended observation during informal interviews) and captured findings in real time [32, 33] |
| 5 | *Community engagement* | Researchers used community engagement to enhance and accelerate participation and support [34–36] |
| 6 | *Social media recruitment* | Recruitment was conducted through popular online social media and social networking websites and apps, such as Facebook, Instagram, and Whatsapp [37, 38] |
| 7 | *Remote data collection* | Data collected remotely by telephone, Whatsapp, and email all reduced data collection time [39] |
| 8 | *Mixed methods data collection* | Mixed methods data collection enabled rapid publication of results and simple graphic presentation of research findings [40, 41] |
| 9 | *Use of existing data sources* | Researchers used existing data bases and known documentary and recorded sources to support or validate analyses [42, 43] |
| 10 | *Digitized data entry and analysis* | Simultaneous data entry on Google Forms, Excel or other software permitted simultaneous data collection, transcription, and preliminary analysis [30] |

ethical approval. The remaining studies (n = 81, 64.3%) did not report ethical review and approval, or indicated that they did not seek it. Some bypassed approval in accordance with the requirements of funders or institutions providing direction. Others used open-source materials for rapid analysis. There was no discussion of expedited review procedures, even though ethics review processes can typically take weeks to months.

## G. Training

Few studies (8%, n = 10) addressed field researcher training to conduct RQM or training content. Such sources involved close collaboration with community-based organizations, local groups, or institutional actors [26, 44, 45].

Trainings were often implemented when sufficient numbers of trained field assistants were not available, typically in LMICs, or where ethnographic methods entailed working in multilingual contexts or in specific dialects [6, 19, 46]. The involvement of national researchers in training and leading research in their own countries frequently could not be determined, with a few exceptions [35, 47].

## H. Research questions addressed by RQMs

The rapid review found that RQMs have been deployed to address specific categories of research questions (see Table 6).

RQMs were frequently used to understand the perceptions, experiences, knowledge, and responses of community members, patients, families, and other social groups to epidemic transmission and control measures [16, 19, 48, 49]. They were also employed to assess the mental health and psychosocial challenges confronting healthcare workers working in high stress situations [28, 50, 51]. They also uncovered social, cultural, and religious factors affecting local populations' willingness and ability to uptake public health control measures, accept public health communications, utilize vaccines, or seek access to emergency services [52]. RQMs identified and shed light on the qualitative conditions confronting vulnerable populations affected by the epidemic. RQMs were employed in communication and media studies to investigate the framing and representation of health issues throughout different media landscapes as well as information distribution and sharing via different media platforms [9, 39]. We were unable to code for these topics because linguistic and technical terms were applied inconsistently and changed across units of analysis.

Certain rapid methods investigated specific and contextually sensitive interpretations of vulnerabilities and resilience in disease outbreak contexts, rather than presuming the nature of vulnerability to better define and gain insight into marginalization or exclusion [18, 26, 53–56].

## I. Funding

Included sources reported funding sources, but did not provide details on funding awards, selection procedures, or task appropriations. Normally, such information is not divulged in publications or gray literature, and costs will vary with the location, timing, and research objectives. Efforts to reduce costs can affect study validity and scope [35, 57].

## J. Community engagement

Nearly half of included sources (44%, n = 55) mentioned community engagement. This engagement included recruitment for focus group discussions [30]); sharing results with study participants [44]; and the recruitment of field assistants for data collection [6]. Rhodes and colleagues' RQM was conducted with a pre-existing partnership of gay, bisexual, and other men

**Table 6. Thematic analysis of types of studies using RQMs in epidemics.**

| | |
|---|---|
| **Populations Researched** | • Healthcare workers<br>• Patients<br>• Patients' families<br>• Public/communities<br>• Vulnerable populations (HIV, migrant, elderly, children, homeless populations)<br>• Stakeholders (governments, NGOs)<br>• Medical/healthcare facilities<br>• Educational institutions |
| **Themes** | • Gender-based themes (gender-based violence, pregnancy, abortion)<br>• Community perceptions:<br> ◦ Knowledge, attitudes, and practices of populations towards vaccines<br> ◦ Local interpretations of the disease<br> ◦ Patient perceptions of healthcare workers, healthcare facilities, healthcare systems, and novel interventions<br> ◦ Reticence/resistance to vaccines, control measures, new technologies, tests, or communications<br> ◦ Effectiveness of the epidemic response<br>• Patient perceptions and experiences<br>• Healthcare worker perceptions and experiences<br>• Mental health and psychosocial well-being of patients, survivors, and healthcare workers<br>• Emergency facilities<br>• Rumours and misinformation<br>• Public health communications<br>• Quality and perceptions of health systems preparedness and response<br>• Safe and dignified burials |
| **Types of Research** | • Assessment<br>• Needs assessment<br>• Evaluation<br>• Systems analysis<br>• Ethnographic reporting<br>• Multi-stakeholder analysis<br>• Guidance and Recommendations<br>• Lessons learned |

who have sex with men; public health department actors, clinics, and HIV organizations [58]. Ferrer-Garcia and colleagues recruited teachers and students to carry out their RQM in an educational institution to address pedagogy during the COVID-19 pandemic in Cuba [59].

## K. Restitution ("follow up")

One-third of included sources reported *restitution*, or "restoring" the RQM data or findings to participants, communities, authorities, or other stakeholders. Early in the COVID-19 pandemic, for instance, Kra and colleagues reported their findings concerning end-of-life in the Intensive care unit of the European Hospital of Marseille (HEM), to the COVID-19 mobile team and the hospital management [60]. In a Delphi study of experts in pediatric dialysis, study researchers shared a manuscript draft with study participants prior to publication [61]. Another RQM that investigated Lebanon's response to COVID-19 at municipal levels, and the pandemic impact on most marginalized urban inhabitants shared results with municipal unions and other stakeholders [27].

### L. Impact

The impact of specific RQMs was of particular interest. Although we initially sought to examine citations as an indicator of impact, the Methods subgroup concluded that this measure was inappropriate for diverse RQM sources. Most studies had transformative *intent*. RQMs often accompanied qualitative or community interventions, or were meant to provide narrative accounts of conditions, experiences, systems, or networks during disease outbreaks. Some studies sought to transform current delivery of services or to offer insights that could reshape institutional understandings of specific problems during an epidemic [27, 35, 62]. Desclaux and Sow, for instance, conducted an RQM of volunteers who followed contact cases during the West African Ebola epidemic, and their recommendations for contact tracing were adopted by the Senegalese Ministry of Health and the Red Cross. Following their RQM examining Zika, pregnancy, and contraception in the Virgin Islands, Brittain and colleagues worked with a marketing company to develop and implement a full-fledged communications campaign (posters, brochures, radio spots, a website, and social media content) on Zika and pregnancy [45]. No RQM studies used quantitative or statistical analytical tools or research design to determine impact.

## IV. Rapid qualitative methods assessment and reporting tool

The Rapid Qualitative Methods Assessment and Reporting Tool is a key result and product of this rapid review, intended to improve RQM study quality in and reporting of research design, implementation, evaluation, dissemination, and review during epidemic emergencies.

Through the findings of our rapid review of RQMs in epidemic emergencies, we identified 13 key criteria in three domains for evaluating RQMs in epidemics and disease outbreaks. The three domains and criteria are as follows: **I. Design and Development** (i. time frame, ii. Training, iii. Applicability to other populations, iv. Applicability to low resource settings, v. community engagement, vi. Available resources, vii. Ethical approvals, viii. Vulnerability, ix. Tool selection); **II. Data Collection and Analysis** (x. concurrent data collection and analysis, xi. Targeted populations and recruitment procedures); **III. Restitution and Dissemination** (xii. Restitution and dissemination of findings, xiii. Impact).

Table 7 summarizes short definitions for these criteria; detailed descriptions of each criterion follow.

These criteria align with the process for the development, implementation, and completion of RQMs.

## V. Discussion

The past decade has seen multiple important interventions concerning RQMs, including a systematic review addressing English-language studies in epidemic emergencies, reflections about the use of RQMs in the COVID-19 pandemic and two handbooks about RQMs more generally [3, 4, 9, 63]. These publications have offered crucial insight into uses, strengths, and limitations of RQMs and provided valuable guidance for researchers seeking to develop and deploy them. The present article builds on and extends these publications, by expanding analyses of RQM features and impacts in French, Spanish, Portuguese, and Mandarin and in new geographical regions (Asia, Latin America), and including less recognized or newly developed methods. Based on this analysis, we elaborated an RQM assessment and reporting instrument to assist multiple users (e.g. funders, researchers, evaluators, peer reviewers) to plan, implement and evaluate RQM studies in epidemic response and to encourage greater transparency in study reporting.

**Table 7. Rapid qualitative methods assessment and reporting tool for epidemic emergencies.**

| Criteria | Definition | Domain |
|---|---|---|
| i. Timeframe | Speed of the study, from team training through data collection, analysis and some type of reporting | I. Design and Development |
| ii. Training | Duration, ease and accessibility of training platforms, complexity of methods needed for team preparation | I. Design and Development |
| iii Applicability to other populations | Whether the method could be repurposed for populations and settings other than those researched | I. Design and Development |
| iv. Applicability to low resource settings | Low resource settings pertain to lower- and middle- income countries, but also to multiple scales in LMIC and high-income countries, from individuals and households to neighbourhoods, cities and regions | I. Design and Development |
| v. Community engagement | defined as participation among study population in any aspect of the development, testing, implementation and analysis of the method, or mobilizing to act on study results | I. Design and Development |
| vi. Available resources | Onsite human and material resources and overall costs of implementing a rapid qualitative assessment are key factors in gauging the feasibility of a specific rapid method | I. Design and Development |
| vii. Ethical approvals | One or more qualified institutions or committees may need to evaluate and approve a rapid qualitative study prior to its implementation | I. Design and Development |
| viii. Vulnerability | How well the methodology illuminates new or complex, interacting processes of marginalization or exclusion, or highlights and new forms of vulnerability | I. Design and Development |
| ix. Tool selection | the choice of adaptable, appropriate tools that can rapidly yield robust data and operational results and be made publicly available during publication. | I. Design and Development |
| x. Simultaneous data collection and analysis | This process involves conducting data collection and analysis contemporaneously, thus allowing the research team to rapidly refine and identify additional questions and to produce rapid operational insights. | II. Data Collection and Analysis |
| xi. Targeted populations and recruitment procedures | Target groups are those whose conditions were being investigated in rapid qualitative assessments or who would be the beneficiaries of interventions resulting from the assessments. Recruitment procedures refer to processes by which participants in the study were approached and integrated into the study. | II. Data Collection and Analysis |
| xii. Restitution and dissemination of findings | Restitution refers to the reporting of findings to participants, communities, stakeholders involved in the study. The term draws heavily from the notion of "restoring", in that researchers give something back to study participants. Dissemination refers to the diverse means by which researchers can report findings rapidly and effectively. | III. Restitution and Dissemination |
| xiii. Impact (actual or potential) | Rapid qualitative method addressed an important issue, used one or more innovative tools, had methodological rigor that would be convincing to non-social scientists, and could transform current delivery of services and/or understandings of the problem. | III. Restitution and Dissemination |

Below we discuss the specific findings that our rapid review identified and reconfigured as 13 key criteria in our RQM assessment and reporting tool.

## Time frame of the study

The study time frame for conception, implementation, data collection, analysis, and producing actionable findings is a defining criterion of quality for an RQM. Eighty-eight percent of research time frames ranged from several days to four months, covering conception to data collection and reporting (but not peer-review publication), with 12% of studies requiring 4–6 months for completion. Of the sources we evaluated, the following effectively reported how they conducted qualitative research within time constraints, obtained sufficient participation to meet demands for methodological rigour, and delivered their analyses to public health actors within public health timeframe demands [6, 26, 35, 56, 62, 64]. In keeping with these findings, the Subgroup concluded that RQMs adhere to a time frame of up to four months, from conception to analysis. Many studies, however, lacked specific information about time frames, notably whether they included only data collection and reporting or encompassed a fuller research process from conception to publication. We do not stipulate a common

standard here because we recognize that RQM researchers work within varied research implementation structures. In our review, certain investigators received funding to undertake discrete research, whereas others could be part of response coordination teams regularly mobilized to collect data.

## Training

Rapid qualitative methodologies require specific skillsets, including the ability to engage people with diverse backgrounds and experiences in conversation; the capacity to ask well-framed, understandable, and probing questions, as well as the skills of careful listening and posing questions to deepen insight into specific situations. Strong skills to observe nuanced or nonverbal communication during community engagement and data collection, to build a rapport with participants, and to conduct reflexive analyses to pinpoint research biases and assumptions are all essential for RQMs. The training of data collectors and other study team members should be reported. All research team members, particularly field researchers, should receive some training in qualitative methodologies, although our included sources rarely commented on training procedures. Even before epidemic outbreaks, long-term investments in qualitative research training were needed to ensure that field and supervisory researchers were prepared to respond when the moment demanded. Training alternatives should be provided in contexts that lack experienced qualitative social scientists. Systematic investments in pre-packaged, asynchronous, virtual, or remote mentoring training systems can ensure that local researchers and national and local stakeholders lead in setting research agendas, developing and implementing both rapid and long-term research, and owning the research process.

## Applicability to more than one group

Standardized protocols, if available, can be used and adapted to local populations to enhance comparability of RQM data and to optimize the speed and efficiency of research design, data collection, and analysis. Such protocols, when structured, were useful in situations where non-social scientists wish to conduct qualitative research, or where social sciences capacity was limited. Actors with local knowledge should inform and lead design, develop, and adapt and pretest pre-existing tools to ensure local linguistic, cultural, and contextual relevance. Volkmann et al. demonstrated that when specific tools were available for consultation and adapted to specific settings, they could ensure rapidity, rigor, and replicability of the study in diverse contexts [56]. During epidemics, sharing RQM protocols through publication, repositories, and other means is crucial for enhancing collaboration and promoting their broader use.

## Applicability to low-resource settings

Rapid qualitative methods must be accessible and feasible for use in settings where resources are limited. LMIC actors should lead in the strategic direction and development/adaptation of rapid methodologies and their implementation for qualitative research in epidemic emergencies. Certain sources employed methods and tools that were especially well-adapted to LMICs [6, 53–56, 65]; and used free and open-source software platforms to facilitate local researcher participation in data access and analysis [14].

Our analysis found that rapid qualitative assessments in low-income countries tended to adopt face-to-face methods, such as formal and informal interviews, focus group discussions and participant observation, to overcome the absence or unreliability of internet or other digital infrastructures. Remote data collection tools (remote interviews, journals, social media, and photo journaling) were useful in some contexts, but not a sufficient part of the rapid qualitative research toolkit in epidemics.

We also noted that although most rapid tools that the Methods Subgroup examined were being used in LMICs, very often studies using these tools were led by 'global health' actors, not local researchers. It is possible that this is a result of publication bias, but our investigation did include grey literature. The implication of local actors in development of the tools was unclear as a result.

## Community engagement

Community engagement before, during, and after a rapid qualitative assessment enhances the quality of rapid research and post-research interventions, in that it responds to the priorities, values, and concerns of the communities in which it takes place [18, 27, 64, 66]. Community engagement ensures that rapid research has a concrete impact. It can enhance the design, acceptability and quality of research, by ensuring that measured outcomes reflect community-identified priorities; identify and minimize internal risks; strengthen informed consent processes through dissemination of information on research goals, risks and benefits, and incorporating local views into consent processes; and empower communities and demonstrate respect, as a goal in itself and as a means of strengthening mutual understanding, trust and credibility between researchers and participants [64, 67–70].

Community participation should be emphasized throughout the entire research process, including setting of research priorities, framing of research questions, and the development and testing of research tool, analysis, restitution, and mobilization for further action.

## Available resources (material and human) and costs

Costs for materials, equipment, software, researcher training costs, travel costs, and participant reimbursement are known barriers to effective and rapid qualitative research in epidemics. In practice, resource availability (including its distribution) can fundamentally affect the scope and effectiveness of an RQM [4, 63]. In emergency situations, for instance, availability of human resources and funding are frequently imbricated and can affect the success of an RQM. This criterion also streamlines the comprehensive assessment of resource adequacy for an RQM, leading to a more informed decision-making process for funders, researchers, and practitioners. These available resources and costs were not reported in our included sources.

## Ethical practice

Ethical practice is fundamental to conducting any research, including rapid qualitative research, and we recognize that in some contexts, certain operational research is not subject to formal ethical approval. Our included sources did mention when ethical approvals were obtained, but did not discuss the procedures for doing so, particularly under urgent conditions.

Collaborations with local authorities and communities can be essential to ensuring that the RQM will be ethically appropriate in the specific local context. RQMs now tend to be led by researchers coming from the same countries where epidemic emergencies occur. In reporting RQMs, there should be a description of ethical practices, and where appropriate, of the approval, duration, and use of accelerated review procedures.

## Vulnerability

In epidemics, RQMs can provide insights into new or hidden forms of vulnerability in epidemic emergencies, which exacerbate inequalities and can push people already facing difficulties into further marginalization. It is critically important to attend to specific, contextually

sensitive vulnerabilities, without presuming in advance which social groups might be marginalized or excluded. An RQM's usefulness in examining vulnerability may vary, depending on context and research question addressed. Further, we found qualitative methods were most illuminating when revealing how and why such vulnerabilities emerge [18, 34, 56].

### Tool selection

Most studies reviewed used diverse data collection approaches. Tool innovations were observed in Honorato and Oliveiro, who combined key informant interviews with analyses of online news sources to suggest policy strategies concerning homeless people for local government officials in Brazil during the COVID-19 pandemic [66]. Future work on RQMs should systematically document further and evaluate such innovation.

### Target groups and recruitment procedures

RQMs should systematically detail and justify recruited participants and recruitment procedures. Included sources frequently reported numbers of recruited participants but did not detail either the procedures or justifications for numbers recruited. Although longer-term studies can rely on "saturation" (data collection until the researchers no longer learn anything new) as a target, RQMs should make clear not only the numbers of participants but specifically why those numbers are sufficient to produce robust results.

Under certain conditions, target populations are not easily accessible [7]. Certain studies were especially effective in reaching populations that are frequently understood as "hard to reach" and particularly in capturing the experiences and emotional needs and responses of these populations [6, 18, 26, 53].

### Simultaneous data collection and analysis

Although only one of our sources explicitly sought to undertake data collection and analysis simultaneously, the Methods Subgroup contended that it could enhance quality and rapidity of an RQM. Rapid qualitative assessments required data collection and analysis to occur simultaneously and iteratively, to ensure that data collection responds over time to new findings [32]. Resources should be devoted to transcription software and/or team so that transcriptions can be done as rapidly possible. During the implementation process of an RQM, research teams scheduled frequent debriefings, wrote analytical memos, adjusted numbers and characteristics of included participants, questions asked and tools used, and triangulated with other types of data. Periodic memos or reports on the bases of these debriefings can help track changing research priorities in response to realities on the ground. They can also provide the basis for rapid reporting to stakeholders. This factor is closely related to cost and resource availability, since simultaneous data collection and analysis requires sufficient numbers of trained people.

### Restitution and dissemination

*Restitution* "restores" findings to research participants, communities (in the largest sense), stakeholders, and authorities. It is one form of dissemination of findings, but it is a form of dissemination that is crucial for impact, ethics, community engagement, data stewardship and ownership, and community empowerment in the research process. Several sources systematically conducted restitutions of their findings, reporting preliminary findings and furnishing reports to authorities and stakeholders on a regular basis [15, 27, 35, 54, 62]. Quick, simple

dissemination, such as developing one-page templates for frequent reporting to stakeholders was useful.

The rapid review found multiple means of disseminating results to stakeholders, authorities, and communities, including rapid periodic reports, public statements, rapid briefings, and public conferences or webinars. Local and international organizations may help to share the research results with participants and local communities. International organizations, for instance, publish reports and summaries of rapid research and publicize their availability through online campaigns.

Peer-reviewed publication should never be a substitute for restitution. Indeed, academic publication may be delayed in a health emergency, so therefore should not be considered the only route to disseminating findings and producing impact; other methods should be employed to engage stakeholders and inform health emergency responses at the appropriate level. Certain research teams have closely collaborated with Ministries of Health from early in the research process to ensure engagement with and rapid uptake of their results [46].

### Impact

The Methods Subgroup defined "Impact" as the consequences that qualitative research had for epidemic policy and practice, and we rejected traditional scholarly standards for impact, notably number of citations or journal rankings. Certain sources [27, 35, 62] achieved impact through engagement with key decision makers. Desclaux, for instance, conducted a rapid qualitative study of volunteers who followed contact cases during the West African Ebola epidemic [62]. Their recommendations were adopted by the Senegalese Ministry of Health and the Red Cross.

## VI. Strengths and limitations

The evaluation was conducted by highly experienced, international expert researchers based in seven different countries. It covered published and grey literatures in five languages; the wide range of epidemic emergencies covered; the coverage of multiple social sciences traditions; a rigorous evaluation process, which included three rounds of inclusion/exclusion evaluation to ensure that publications were appropriately categorized. A language team member, a member of the subgroup, and 2–3 members of the Methods Subgroup read each included source for adherence to inclusion criteria and for quality. In developing our evaluation of RQMs, we also consulted other checklists for reporting qualitative research from COREQ and Equator (Enhancing the Quality and Transparency Of health Research) [71].

Our rapid evidence review has some limitations. First, the team made great efforts to include grey literature, as well as summaries of ongoing studies. Nevertheless, given the compressed time frame of the evidence review, we likely missed some relevant published sources, although our included sources were disproportionately published in peer-reviewed journals. In addition, we may not have had access to rapid research that focused dissemination on the frontline response. Finally, we could not include sources in all relevant languages and did not include Spanish and Portuguese publications outside of South/Central America. Future rapid evidence reviews should make efforts to include a wide scope of languages.

## VII. Conclusion

Strengthening the role of social science in disease outbreaks will require more attention to the standards that we use to evaluate and report RQM research in epidemics, attending to the context of implementation and the merits of the analyses generated. This study demonstrates a clear need for structured approaches to evaluate RQMs in disease outbreaks. The Methods

Subgroup, comprised of social scientists experienced in outbreak response from every world region, identified and agreed upon three domains and 13 criteria by which RQMs should be reported and evaluated. The quality of the research and the practicality and ethics of conducting such research during public health emergencies was evaluated. The RQM Assessment and Reporting Tool represents key priorities in qualitative social science and public health research methods and is intended to align with operational research needs in disease outbreak situations. When applied to RQM in public health emergencies, the Assessment and Reporting Tool will serve funders, public health workers, researchers, coordinators, and policy makers by making RQMs more predictable, transparent, and reliable. This, in turn, will further advance the WHO's aim of strengthening the role of social science evidence in public health emergencies.

## Supporting information

**S1 File. Search Strings in all languages.**
(DOCX)

**S2 File. First round data extraction template for rapid qualitative methods.**
(DOCX)

**S3 File. Second round coding of rapid qualitative methods.**
(DOCX)

**S4 File. Language-specific data extraction flowcharts and notes.**
(DOCX)

**S5 File. All included sources, with descriptions.**
(XLSX)

**S6 File. PRISMA checklist.**
(PDF)

## Acknowledgments

We are grateful to members of the COVID-19 Social Sciences Research Roadmap, as well as João Rangel de Almeida and Dayo Spencer-Walters for their feedback on this study. We also thank Anthony Billaud, Renata Gabriela Cortez Gómez, Ximena Pamela Díaz Bermúdez, and Rubén Muñoz Martínez for their important contributions to this study. Finally, we thank A. David Napier for his suggestions during the planning stages of the study.

## Author Contributions

**Conceptualization:** Dong Dong, Gustavo Corrêa Matta, EK Yeoh, Nina Gobat, Tamara Giles-Vernick.

**Data curation:** Dong Dong, Gustavo Corrêa Matta, Tamara Giles-Vernick.

**Formal analysis:** Dong Dong, Sharon Abramowitz, Gustavo Corrêa Matta, Arlinda B. Moreno, Elysée Nouvet, Jeni Stolow, Caitlin Pilbeam, Shelley Lees, Tamara Giles-Vernick.

**Funding acquisition:** Tamara Giles-Vernick.

**Investigation:** Arlinda B. Moreno.

**Methodology:** Dong Dong, Gustavo Corrêa Matta, Arlinda B. Moreno, Elysée Nouvet, Jeni Stolow, Caitlin Pilbeam, Shelley Lees, EK Yeoh, Nina Gobat, Tamara Giles-Vernick.

**Supervision:** Nina Gobat, Tamara Giles-Vernick.

**Validation:** Dong Dong, Tamara Giles-Vernick.

**Visualization:** Sharon Abramowitz.

**Writing – original draft:** Dong Dong, Sharon Abramowitz, Jeni Stolow, Tamara Giles-Vernick.

**Writing – review & editing:** Dong Dong, Sharon Abramowitz, Gustavo Corrêa Matta, Arlinda B. Moreno, Elysée Nouvet, Jeni Stolow, Caitlin Pilbeam, Shelley Lees, EK Yeoh, Nina Gobat, Tamara Giles-Vernick.

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
