## [Decision Letter · Decision Letter 0]

18 Jun 2023

PGPH-D-23-00594

A Matrix for Reporting and Evaluating Rapid Qualitative Methods for Epidemic Response as the Outcome of a Rapid Review and Expert Consultation

Dear Dr. Giles-Vernick,

Thank you for submitting your manuscript to PLOS Global Public Health. After careful consideration, we feel that it has merit but does not fully meet PLOS Global Public Health’s publication criteria as it currently stands. Therefore, we invite you to submit a revised version of the manuscript that addresses the points raised during the review process.

This was a strong manuscript overall and will be a valuable contribution to the literature on qualitative research methods. We do, however, encourage you to address the points raised by the reviewers -- particularly Reviewer 2's feedback regarding the structure and organization of the manuscript.

We look forward to receiving your revised manuscript.

Kind regards,

Sanjana Ravi, PhD, MPH

Academic Editor

Journal Requirements:

1. Please send a completed 'Competing Interests' statement, including any COIs declared by your co-authors. If you have no competing interests to declare, please state "The authors have declared that no competing interests exist". Otherwise please declare all competing interests beginning with the statement "I have read the journal's policy and the authors of this manuscript have the following competing interests:"

2. Please provide separate figure files in .tif or .eps format.

Additional Editor Comments (if provided):

Reviewers' comments:

Reviewer's Responses to Questions

**Comments to the Author**

1. Does this manuscript meet PLOS Global Public Health’s publication criteria? Is the manuscript technically sound, and do the data support the conclusions? The manuscript must describe methodologically and ethically rigorous research with conclusions that are appropriately drawn based on the data presented.

Reviewer #1: Yes

Reviewer #2: Yes

2. Has the statistical analysis been performed appropriately and rigorously?

Reviewer #1: Yes

Reviewer #2: N/A

3. Have the authors made all data underlying the findings in their manuscript fully available (please refer to the Data Availability Statement at the start of the manuscript PDF file)?

Reviewer #1: Yes

Reviewer #2: Yes

4. Is the manuscript presented in an intelligible fashion and written in standard English?

Reviewer #1: Yes

Reviewer #2: Yes

5. Review Comments to the Author

Reviewer #1: Very interesting paper with a clear methodology. The study selection process included in the review is well documented. The conclusion is also appropriate. I learned a lot from reading this article. It was a privilege to be among the first readers of this rich and scientifically valuable work. Congratulations to all the authors.

A total of 114 studies were included out of the 8,405 initially identified. Indeed, the exclusion criteria were well described and the study selection flow chart is clear.

Question 1 (Minor): Does the high proportion of articles eliminated (98.6%) suggest another study limitation, given that the studies excluded could reveal a greater diversity of objectives and methods used when conducting qualitative researchs during outbreak ?

Question 2: The study selection flow chart shows 12 studies that were added at the end of the selection process (114 + 12 = 126). This suggests that these 12 studies were not subjected to the same selection criteria as the other studies. What was so different about these 12 that they were not evaluated like the other studies? What criteria were they subjected to before inclusion in the study ? Maybe a few details in the methodology might spare readers from asking this question.

Question 3 (Minor): Krippendoff's α coefficient has been reported for inter-encoder agreement. However, nothing was said about how it was calculated. Specifically, was any software used ? Would it be useful to add a table (in the appendix) showing this calculation?

Question 4: The research team has suggested a 13-criteria matrix for qualitative studies during emergency context. a) Given their importance and specificity, including the difficulty of collecting funding data, shouldn't human resources and funding criteria be presented separately? b) Concerning the second objective of the study (to propose a matrix for notifying and evaluating rapid qualitative studies in emergency situations), wouldn't it be more comprehensive if the research team suggested a scale score for each criteria and for the 13 criteria as a whole? In its current state, this matrix seems to me to be useful only for notification. But if studies are also to be evaluated, a scale should be suggested.

Reviewer #2: Overall comments

This is an excellent paper, and I am delighted to have had the opportunity to review it. It gave me a lot of food for thought and will improve my own approaches to RQM. My comments are largely around clarity and structure, as the content itself is very strong. The inclusion of multiple languages in the review is a particular asset of this work.

My main editorial point is that I feel the paper is missing the aspect of a discussion that typically contextualises the findings in the literature and really underlines the contribution. The focus on the evaluation matrix is important, but I was left wanting a reflection on how the development of these criteria moves the field forward relative to what we knew before. You make a lot of novel points and I would like to see these shine through more. Something that I was consistently unsure about across the paper was about duration of studies – clarity throughout about whether the end of a study was reporting/ end of fieldwork would be very helpful.

Introduction

- On line 79 you say this expands on prior research, but it would be useful to have more about what this prior research found. I suggest you add more detail on what is already known in the existing methodological literature (e.g. a summary of Johnson and Vindrola-Padros’ 2017 review).

Methods

- Line 117 – what was the previously published study that you used to replicate the search terms? Could you say more about what it was not possible to replicate the resource extraction?

- Why limit the Spanish/Portuguese sources to Latin American countries – why not do a global search to capture sources published in Spanish or Portuguese from Europe as well? You mention this as a limitation, but it would be useful to acknowledge it in the methods as well.

- Could do with a bit more detail in the section on coding, analysis and synthesis – the process seems robust but it isn’t 100% clear how you came to the themes. It would help to know more about how you approach the initial data extraction – i.e. how did the core methods subgroup come up with the initial coding template – and how did you decide the themes through group discussions? Basically, I’d like to really understand how you moved from the literature review through the criteria of RQM quality.

- Can you clarify the ‘duration of research’ table - Does ‘duration of research’ relate to the total study time (i.e. from conception to reporting) or to the duration of fieldwork?

Results

- 204 – can you clarify ‘completed’ – does this mean reported/published?

- 249 – when you say tools, do you mean the specific method (i.e interview), the way of doing the method (i.e. telephone/video interview) or the resources informing the data collection (i.e. topic guide)?

- 257 – why ‘possibly’ in rural regions? Because this wasn’t clearly reported?

- In section ‘C- analytical approaches’ the 2nd and 3rd paragraph repeat some of the same information. This could be condensed.

- The box ’10 observed strategies for making rapid qualitative methods ‘rapid’’ only has 9 points

Evaluation matrix

- It is a useful summary of different aspects of RQMs, but I think it would be helpful to state its purpose more clearly. I interpreted that the table is meant as a way to a) inform design of quality studies using RQMs and b) to assess the quality of other RQM studies. I wasn’t 100% sure though, so be really specific about this.

- This section could also from some signposting, as initially I thought it was part of the results. Perhaps the way to address this is to move this section entirely into the discussion.

- As a further point – is it a matrix? Earlier in the paper you refer to it as a framework, but I’m not sure if is that either. It is a set of criteria that indicate quality. Think on this as a writing team and come up with a consistent way to describe the table.

Discussion

- As mentioned above, I would recommend bringing the evaluation matrix into this section, with a clear introduction signposting that this is the research team’s interpretation and repurposing of the findings of the review (linking back to what I said re: methods – how did you get to this point through your analysis?)

- 397 – could you clarify if the timeframe of 4 months for conception to analysis also includes reporting?

- In the discussion of training, worth reflecting on skills needed for collecting/analysing data other than interviews/observations.

- As above, in the section on ‘applicability to more than one group’ could you clarify what you mean by ‘tools’?

- I agree with the point about the usefulness of RQM in epidemics for exploring vulnerability, but I’m not sure this would be considered a methodological marker of quality for all studies, as it surely depends in part on the questions being asked. Is there a way of caveating this point slightly to reflect this?

- Lines 475-476 – I think this is a comment from someone in the writing team, and I agree with it!

- Line 488 – this is first mention of ‘the expert committee’, who was this? The writing group? Someone else? If so this should be clearer in the methods section.

- Line 516 – ‘Desclaux’ is named rather than presented as a numbered reference

- I would also expect a discussion to include a return to the aims of the paper and to summarise the contribution of the paper relative to the existing literature.

General points

- I was interested to see the variation in duration of research across studies (how did 18 studies complete the work in less than a week!?). Did you notice any different in reporting/ methodological congruence between the shorted and longest rapid studies? Does there seem to be an ‘optimum’ duration?

- Was there anything about emotional impact on researchers of rapid research? It strikes me that there are different pressures involved in conducting RQM, potentially, e.g. researchers who may end up doing multiple emotionally challenging interview in rapid succession. If not mentioned in the literature, something to reflect in in the discussion.

- For me a really key finding is that RQM can both be good at accessing ‘hard to reach’ groups and ‘low-hanging fruit’ groups, i.e. professional stakeholders. So this suggests that the positionality of the research team is a really important part of RQM, as who they have immediate access to will likely form the study participants. Again, if not discussed explicitly in the literature perhaps something for the discussion.

Typos

Line 139 – in exchanges with funding?

Line 165 – a key limitation is the lack…

Line 306 – others used open-source materials for analysis

Line 493 – not sure what this means – ‘wrote analytical memos to adjust those included’

Consistency of spelling – gray/grey

Thank you for doing this important work, and for the opportunity to review your paper. I look forward to your revisions and eventually seeing it in print. I think it will be a very useful resource for anyone considering RQM.

6. PLOS authors have the option to publish the peer review history of their article (what does this mean?). If published, this will include your full peer review and any attached files.

**Do you want your identity to be public for this peer review?** For information about this choice, including consent withdrawal, please see our Privacy Policy.

Reviewer #1: **Yes: **Gabriel Kyomba Kalombe

Reviewer #2: **Yes: **Anna Dowrick

---

## [Editor Report · Decision Letter 1]

3 Aug 2023

A Rapid Qualitative Methods Assessment and Reporting Tool for Epidemic Response as the Outcome of a Rapid Review and Expert Consultation

PGPH-D-23-00594R1

Dear Dr Giles-Vernick,

We are pleased to inform you that your manuscript 'A Rapid Qualitative Methods Assessment and Reporting Tool for Epidemic Response as the Outcome of a Rapid Review and Expert Consultation' has been provisionally accepted for publication in PLOS Global Public Health.

Best regards,

Sanjana Ravi, PhD, MPH

Academic Editor